# High-Efficiency Fabrication of Geometric Phase Elements by Femtosecond-Laser Direct Writing

**DOI:** 10.3390/nano10091737

**Published:** 2020-09-01

**Authors:** Shuai Xu, Hua Fan, Si-Jia Xu, Zhen-Ze Li, Yuhao Lei, Lei Wang, Jun-Feng Song

**Affiliations:** 1State Key Laboratory of Integrated Optoelectronics, College of Electronic Science and Engineering, Jilin University, Changchun 130012, China; nicholas4125624@163.com (S.X.); sjxu18@mails.jlu.edu.cn (S.-J.X.); zhenze_lee@163.com (Z.-Z.L.); songjf@jlu.edu.cn (J.-F.S.); 2State Key Laboratory of Precision Measurement Technology and Instruments, Department of Precision Instrument, Tsinghua University, Beijing 100084, China; fanhua17@mails.jlu.edu.cn; 3Optoelectronics Research Centre, University of Southampton, Southampton SO17 1BJ, UK; Yuhao.Lei@soton.ac.uk; 4Peng Cheng Laboratory, Shenzhen 518000, China

**Keywords:** nanofabrication, geometric phase elements, femtosecond laser, annealing technology

## Abstract

The nanoresolution of geometric phase elements for visible wavelengths calls for a flexible technology with high throughout and free from vacuum. In this article, we propose a high-efficiency and simple manufacturing method for the fabrication of geometric phase elements with femtosecond–laser direct writing (FsLDW) and thermal annealing by combining the advantages of high-efficiency processing and thermal smoothing effect. By using a femtosecond laser at a wavelength of 343 nm and a circular polarization, free-form nanogratings with a period of 300 nm and 170-nm-wide grooves were obtained in 50 s by laser direct ablation at a speed of 5 mm/s in a non-vacuum environment. After fine-tuning through a hot-annealing process, the surface morphology of the geometric phase element was clearly improved. With this technology, we fabricated blazed gratings, metasurface lens, vortex Q-plates and “M” holograms and confirmed the design performance by analyzing their phases at the wavelength of 808 nm. The efficiency and capabilities of our proposed method can pave the possible way to fabricate geometric phase elements with essentially low loss, high-temperature resistance, high phase gradients and novel polarization functionality for potentially wide applications.

## 1. Introduction

For decades, micro-optical elements have drawn great attention for their advantages of miniaturization and easy integration [1,2,3,4,5,6]. By tailoring the profiles and the corresponding optical path differences—named dynamic phase—various refractive/diffractive optical elements have been realized for micro-optics, microelectronics and biomedical sciences [5,6,7]. However, limited by nanofabrication techniques for high resolution (at least λ/5) and high surface smoothness, it is struggling with complex 3D profiles or the discrete elements at the microscale. Pancharatnam–Berry (PB) phase elements—or geometric phase elements—have emerged as an option [8,9,10,11,12]. They generate continuous phase variation by tuning the direction of the nanogratings, but without changing their thickness [10,11,12]. As a result, the volumes of the optical elements based on geometric phase can be reduced dramatically and the diffractive efficiency can reach extremely high levels, which is highly desirable in integrated optics [13,14,15]. Erez Hasman et al. initially proposed a polarization-dependent 10.6-µm lens based on geometric phase by using computer-designed phase distribution and optical lithography for fabricating diffraction gratings with different directions [15]. Chen et al. fabricated the designed a three-foci metasurface lens on a 40-nm-gold-film-coated glass substrate; electron-beam lithography (EBL) and a lift-off process were used [9].

Generally, the features of gratings for geometric phase elements are subwavelength, indicating that structures hundreds of nanometers wide should be targeted for visible wavelengths. Although various high-resolution nanostructures have been demonstrated by photolithography or electron beam lithography (EBL), more flexible technologies with high throughout and free of vacuum are highly anticipated [16,17,18]. Recently, femtosecond laser direct ablation has come into focus due to its contactless and high efficiency of removing material [6,19,20,21]. High resolution of tens of nanometers has been realized by tightly focused laser beams on transparent materials and semiconductor film through nonlinear absorption and localized light enhancement [22]. However, due to the ultrafast fierce interaction between the femtosecond laser and large amounts of electrons on metal surface, heat-induced sublimated materials flew out and then fell down on the surface, resulting in a jagged nanogroove and randomly distributed nanospheres [23,24]. In this instance, the light will localize and enhance around these nanostructures—instead of transporting smoothly in the designed pattern. The performance of geometric phase elements is unsatisfactory unless material sputtering and the thermal effect are solved.

Herein, we propose a simple and rapid manufacturing method for the fabrication of geometric phase components with femtosecond–laser direct writing (FsLDW) and thermal annealing by combining the advantages of high-efficiency processing and the thermal smoothing effect. The final optical components have better optical performances than those without thermal annealing at a visible wavelength (633 nm). Various geometric phase elements such as blazed gratings, metasurface lens, vortex Q-plate and “M” holograms were demonstrated.

## 2. Materials and Methods

A Yb:KGW solid-state femtosecond laser equipped with a third-harmonic generator (Pharos Light Conversion, Ltd., Vilnius, Lithuania) was operated for FsLDW at a central wavelength of 343 nm, a pulse duration of 300 fs and a repetition rate of 200 kHz. To achieve fast processing, a scanning galvanometer (Sunny Technology, Ltd., Beijing, China) and a 4-focal optical setup were used for focus-shifting at high frequencies. An optoelectronic modulator served as a quick-response shutter was integrated in the laser system to block the beam during the focus shifting. In order to realize high resolution, an objective lens with high numeric aperture (Nikon, 20×, NA 0.75, 60% transmission at 343 nm) was used after a 3× beam expander system improving the diameter of the spot slightly larger than the entrance pupil of the objective. All the pulse energy used in the experiments are 4 nJ. To avoid the asymmetric line width induced by scanning directions vs polarizations, a circular polarization converted from linear polarization was implemented by a Glan prism and a λ/4-wave plate. In the experiment, 50-nm-thick gold films evaporated on the fused silica (Jinlong Photoelectric, Ltd., Changchun, China) were chosen for the fabrication of geometric phase elements. A tube furnace (special Cinite, Beijing, China) was implemented to anneal the gold nanoparticles on the film at different temperature and sustain time in vacuum.

The schematic two-step manufacturing procedure of the geometric phase elements is shown in Figure 1. First, nanogrooves on gold film were written by FsLDW and nanogratings with different orientations were constructed. Due to the electron collision by inverse Bremsstrahlung absorption and the subsequent electron–lattice interaction, a sharp temperature increase resulted in material ablation around the focus center and the formation of nanogrooves. Benefitting from the circular polarization, the localized light field enhanced by the preformed structure’s feedback was symmetric, allowing a uniform nanogroove width in arbitrary scanning directions. However, the ablation and the thermal diffusion was so strong that the edge of the nanogratings was jagged and many resolidified droplets/nanoparticles appeared, which would disturb the geometric phase by introducing additional localized light enhancement. To solve this problem, thermal annealing was used to first melt the particles sputtered on the gold film, and then smooth the morphologies of nanogrooves. Generally, it takes 50 s for the fabrication of any designed pattern in the scale of 200 µm × 200 µm by FsLDW and 1 h for the subsequent annealing process. Most importantly, the maskless and vacuum-free capabilities make the approach highly efficient for the production of geometric phase elements with superior quality.

## 3. Results and Discussion

The fabricated nanogratings with a high production efficiency are shown in Figure 2. With the FsLDW, nanogratings with different orientations were constructed on the gold film (Figure 2a). The width of nanogroove of a period was less than 400 nm, which was much lower than the spot size focused by the objective lens ($D=\;\frac{1.22\lambda }{\mathrm{NA}}=558\;\mathrm{nm}\;$). As we mentioned, grooves with sharp jagged edges were observed after FsLDW. Moreover, many gold particles generated by the ablation and thermal diffusion were observed resolidification on both sided of the processing area after laser writing. Some gold particles (cluster or debris) even emerged into a much bigger one when falling on the gold film. Then, we placed the laser-processed gold films on the glass substrate in the middle of the annealing furnace. The annealing heating rate, temperature and sustain time were set by the control panel of the equipment. The structures after annealing are shown in Figure 2c–f. Both annealing temperature and time were optimized. As shown in Figure 2c,d, there was little change for the nanostructures after annealing the gold film at 400 °C for 20 min and at 600 °C for 30 min, respectively. As the temperature increased, the floral and granular nanostructures gradually decreased and the morphology of laser machining area became more regular at 750 °C, 40 min (Figure 2e). Owing to the different thermal properties between bulk material and nanostructured material, the melting point of the nanostructured surface was much lower than 1064 °C [25,26]. The adjustment of an external atomic structure mainly started from 723 °C, where liquefaction layer was formed on the surface with the increase of annealing temperature. When the annealing temperature reached 817 °C, the whole particle melted and submerged into the surface of the bulk gold film (Figure 2f). Whereas the silica substrate had a certain durability at the melting point of gold nanoparticles, the gold cluster or debris resided on the silica just fused into regular nanogratings at 830 °C, 60 min [25]. Due to the surface tension, the adhesion between the gold nanoparticles and the glass substrate was weakened. Gold nanoparticles displaced by the influence of surface energy and surface tension provided a force for the dispersed smaller nanoparticles to be merged into the nanostructured gold film. The density of nanoparticles in the channels of the processing area was significantly reduced, and the edges of the grooves became more regular. This indicated that the nanoparticles aggregated after thermal annealing. Additionally, the periods changed a little bit while the ratio of the ablated region became larger as the temperature and time increased (Figure 2).

The most primary geometric phase element acted as a blazed diffraction raster and had the ability to steer a light beam into several diffracted orders. The design of subwavelength periodic structure orientations can introduce changes of phase retardation. As shown in Figure 3a, the gratings consisted of a four nanoslit array forming four segments. Each segment had identical slit widths and periods. In order to comply with the fourth-order effective medium theory and have optical characteristics, the pitch of the nanograting must be smaller than the incident light wavelength (λ). The design based on a universal strategy can be utilized to fabricate more complicated optical elements. The Fourier optics to determine the phase profile can produce a desired field distribution in the far field. Blazed gratings are featured with a simple phase profile linear with their positions, which either increases or decreases by 2π across one period. The discretized profile for a grating with a period width of 700 nm and 250 nm grooves is shown in Figure 3b. The phase profile was accomplished by gold films with nanobeam wave plates at space-variant orientations.

Diffraction patterns of the geometric phase blazed grating are observed, as shown in in Figure 3d. For an incident right circularly polarized (RCP) beam at a wavelength of 808 nm, part of the light changed handedness to left circularly polarized (LCP) and experienced a phase-pickup equal to 2π after propagating through the geometric phase. The phase profile experienced by RCP or LCP light remained the same. It showed the measured and predicted diffraction efficiency for the first diffracted order at different multilevel geometric phase diffractive optical elements (DOE). The excellent agreement between the experimental results and the predicted efficiency confirmed the expected multilevel discrete binary phase.

The optical elements based on geometric phase have a great potential in integrated optics due to its compact volume and high-power duration by introducing complex space-variation fast axis of waveplate. When light passes through the geometric phase elements, it is easy to calculate the changes of the geometric phase by Jones matrix method. Consuming an RCP wave ($\left|E_{\mathrm{in}}\right.\rangle$) incidents on the geometric phase element with high transmittance, the output electric field ($\left|E_{\mathrm{out}}\right.\rangle$) can be written as follows [15]:(1)$$\left|E_{\mathrm{out}}\right.\rangle =\sqrt{\eta_{\mathrm{in}}}\left|E_{\mathrm{in}}\right.\rangle +\sqrt{\eta_{L}}e^{-i2\theta \left(x,y\right)}\left|L\right.\rangle$$
(2)ηin= 121+eiϕ2 and ηL= 121−eiϕ2
where
$\sqrt{\eta_{\mathrm{in}}}$ and $\sqrt{\eta_{L}}$ are the polarization coupling efficiencies depended on the nature of materials and the shape of nanogratings. ϕ and θ are the dynamic phase and direction of local sub-wavelength nanogratings, respectively. There are two items for $\left|E_{\mathrm{out}}\right.\rangle$, when the ϕ is unequal to π. One is the component of original electric field with right-hand circular polarization, another is the one with opposite to former circularly polarized electric field which is modified by phase changes. This local phase changes make it convenient to fabricate the DOE using FsLDW with a fixed retardance but different orientations of subwavelength nanogratings. Aiming to better characterize the optical performance of geometric phase DOE, an analyzer after the geometric phase DOE was used to eliminate the influence of unmodulated incident light, which was composed of a linearly polarized waveplate and a quarter-wave plate with a 45-degree angle between two optical axes.

In order to demonstrate the high resolution of the fabrication method and smooth the surface, a geometric phase DOE with spherical phase distribution was fabricated on gold film by FsLDW, which can be described as follows [8]:(3)$$\varphi \left(\rho \right)=\;\frac{\pi \rho^{2}}{\lambda f}$$
where ρ is the radius of the polar coordinates, *λ* is the working wavelength of the geometric phase DOE, *f* is the designed focus length (100 μm).

We made a multilevel geometric phase diffractive focusing lens which had a multilevel discrete spherical phase function with discrete levels N = 4, 8 (see Figure 4a–c). The diameter of the geometric phase DOE was 300 nm with 500-nm nanograting period, and the designed minimum line width and length were 150 nm and 300 nm, respectively.

As shown in Figure 4d, the single pulse energy was 4 nJ for ablation of the gold film. Based on the point-to-point high-speed scanning (5 mm/s) strategy, when using a high repetition frequency of 200 kHz, the substrate will not be damaged so that a uniform line width is obtained, and geometric phase DOE can be manufactured rapidly within one minute. The surface of the structure was smooth and large gold particles and debris were removed after thermal annealing. The period of the nanograting was about 470 nm and it was consistent with the design period. The laser beam performance for focusing and imaging is shown in Figure 4e.

Figure 4f shows the intensity distribution at the focus plane of the geometric phase DOE before and after annealing. Regarding intensity, it was obvious that the intensity was much larger by annealing than that just by ablated. The cross-section of the intensity mapping illustrated near 100% enhancement for geometric phase DOE after thermal annealing, which was attributed to the higher quality of nanogratings surface and reduction of the gold particles.

A series of thin Q-plates (as shown in Figure 5b were fabricated with optical vortices q = 3 topological charge. The fabricated Q-plates consisted of 16 grating segments, and an orthometric linear analyze with polarizer can be used to visualize the property of nanogratings in optical elements. In Figure 5c, we can find out that the structure of each segment is very similar, and the same ablation quality proves the uniformity of the side lobes. In Figure 5f, we designed and fabricated a Fourier hologram element operating at a wavelength of 808 nm. The far-field intensity image shown in Figure 5h is an “M”, which was in consistent with the designed pattern. Meanwhile, the structure of each segment is very similar, and the same ablation mode quality proves the uniformity of the segments. As mentioned above, we have described and verified a novel processing method that can produce geometric phase thin elements as designed with nanometer precision. From the results, we noticed that the phase profile of almost any object can now be reflected as a smooth change of the homogeneous optical axis in the film, which can be used as an efficient hologram with unique polarization sensitivity.

## 4. Conclusions

We have proposed a highly efficient and simple method by combining the femtosecond–laser direct writing (FsLDW) and annealing process for manufacturing of geometric phase optical thin elements in a non-vacuum environment. By using a femtosecond laser at a wavelength of 343 nm and a circular polarization, free-form nanogratings with a period of 300 nm and 170-nm-wide nanogrooves were obtained by laser direct ablation at a speed of 5 mm/s in the air. After fine-tuning structures through a annealing process, the surface morphology was improved obviously by removing the debris and smoothing the nanogroove edges under a temperature lower than the melting point of bulk material. With these two steps, blazed gratings, high-quality geometric phase lens, Q-plates of excellent modal purity, as well as remarkably holograms with arbitrary wavefronts were realized in less than 50 s by laser fabrication and 1 h by annealing. Our proposed method is of highly efficient and vacuum free, providing the options for the future wide applications of geometric phase elements for sensing applications, disposable products and verification designs.

## Figures and Tables

**Figure 1 nanomaterials-10-01737-f001:**
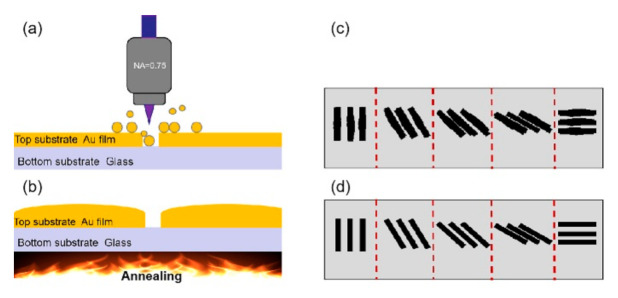
Schematic manufacturing flow of the geometric phase element on gold film by femtosecond–laser direct writing (**a**) and annealing (**b**); (**c**,**d**) schematic nanostructure before and after annealing.

**Figure 2 nanomaterials-10-01737-f002:**
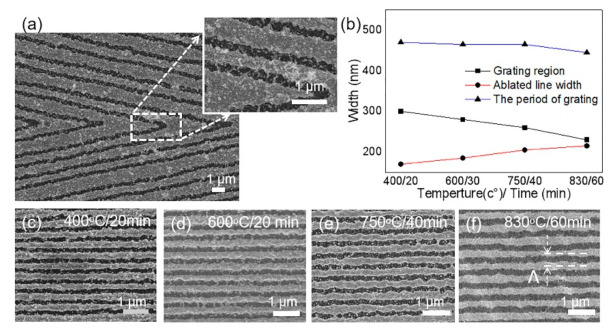
Femtosecond-laser direct ablation of blazed grating at different annealing temperatures and time. (**a**) SEM image of the fabricated dielectric gradient optical elements without annealing; (**b**) ablated line width and the periods of grating vs temperatures and time and their corresponding SEM images (**c**–**f**). Scale bars denote 1 µm.

**Figure 3 nanomaterials-10-01737-f003:**
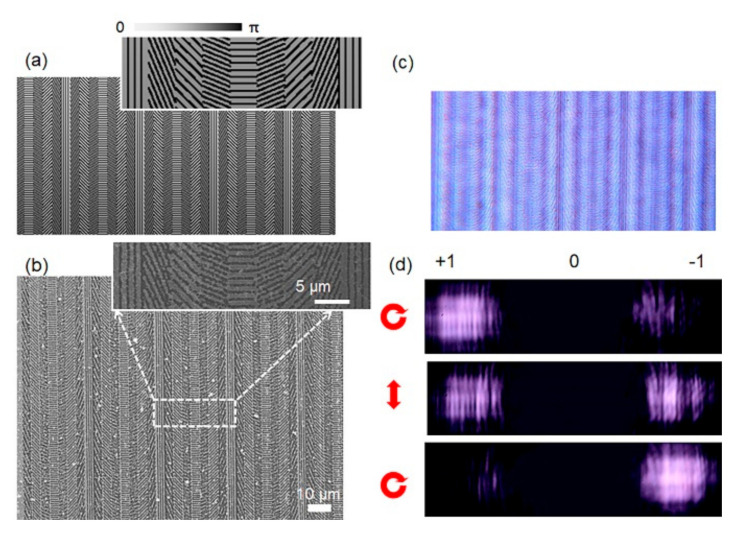
Geometric phase blazed grating. (**a**) Schematic diagram of a geometric phase served as a blazed grating; (**b**) SEM image of the fabricated grating; (**c**) optical microscope image of grating. Dark stripes in these images correspond to locations where the optic axis is aligned parallel or orthogonal to one of the polarizers; (**d**) measured diffraction patterns from the geometric phase under illumination with right circular polarization (top), linear polarization (middle) and left circular polarization (bottom).

**Figure 4 nanomaterials-10-01737-f004:**
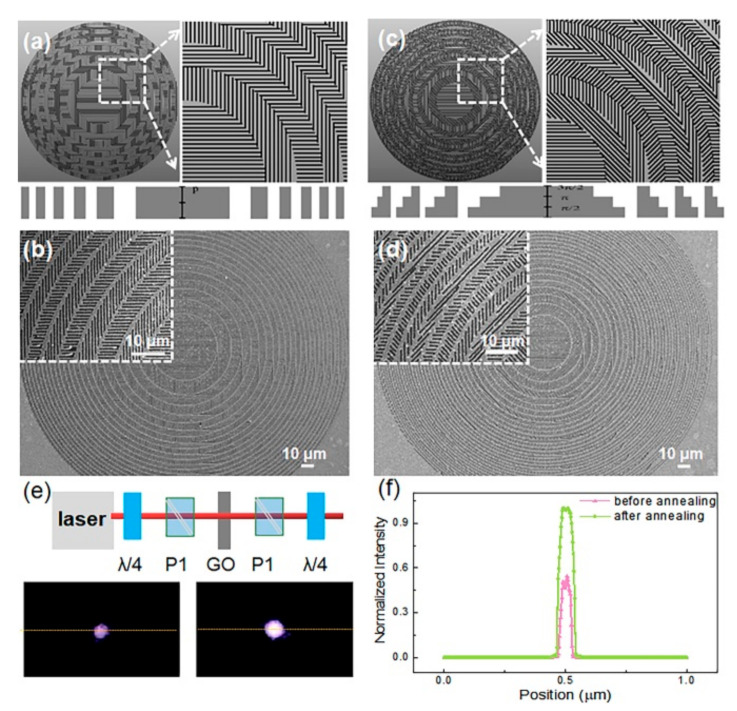
(**a**,**c**) Schematic diagram of the dielectric gradient metasurface lens with discrete level 4 and 8, respectively; (**b**,**d**) SEM image of the lens; (**e**) optical characterization and the intensity pattern at the focus before and after the thermal annealing experiment; (**f**) cross-sectional intensity profile at the focal plane along the dashed line in (**e**). All scale bars denote 10 μm.

**Figure 5 nanomaterials-10-01737-f005:**
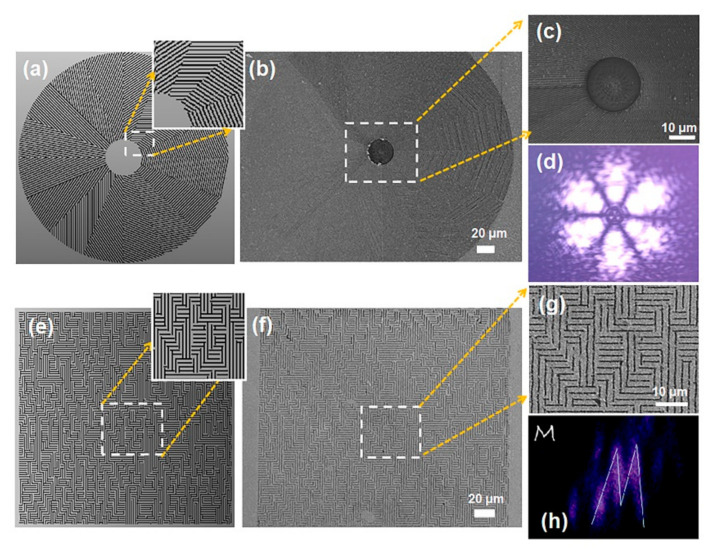
Femtosecond-laser direct writing (FsLDW) of dielectric gradient metasurface optical elements. The designed (**a**) and experimental (**b**,**c**) vortex phase Q-plate. (**d**) Intensity readout of Q-plates with linearly polarized beam; The designed (**e**) and experimental (**f**,**g**) geometric-phase Fourier holograms for a letter of “M”. (**h**) the Fourier hologram element of a letter “M” illuminated by a laser at a wavelength of 808 nm.

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
