# Peer review of "High-Efficiency Fabrication of Geometric Phase Elements by Femtosecond-Laser Direct Writing"

_nanomaterials, 2020, doi:10.3390/nano10091737_

Round 1

Reviewer 1 Report

This paper represents the manufacturing method for the fabrication of geometric phase components with femtosecond laser direct writing and thermal annealing by combining the advantages of high-efficient processing and thermal smoothing effect. Proficiency of this technique is illustrated with fabrication of the some geometric phase elements, for example, metasurface lens and vortex q-plate.

The paper contains results that are of interest for experts in area of the nanophotonics. I believe the manuscript is suitable for publication.

Author Response

Thanks for the overall comments.

Reviewer 2 Report

The authors have done interesting work but it lacks significance and novelty of the work which is basic requirement for research article. I have gone through the manuscript and some concerns for me are listed as follow:

-English language should be checked thoroughly.

-Numbers and their units should contain one space gap between them, so make homogenous and correct the writing.

-The significance and novelty of the work should be clearly mentioned in the abstract, introduction and conclusion sections of the manuscript.

- Insertion of reference citation should be before ending the sentence or ‘full stop’.

-Abbreviations should be defined at first appearance and properly written in same way throughout the manuscript. For eg. FsLDW and FsDLW.

-Fig. 2 caption and the pictures are not clear. Better to show more clear figures along with clear labeling.

-Page 5, define all the variables used in eqn 3.

Author Response

Remarks/Questions

  1. English language should be checked thoroughly.
  2. Numbers and their units should contain one space gap between them, so make homogenous and correct the writing.
  3. Insertion of reference citation should be before ending the sentence or ‘full stop’.

Response to 1 and 3: Thanks. We have revised the article following the comments.

  1. The significance and novelty of the work should be clearly mentioned in the abstract, introduction and conclusion sections of the manuscript.

Response: Thanks. We have revised. The nanoresolution of geometric phase elements for visible wavelengths call for a flexible technology with high throughout and free of vacuum. In this article, we propose a high-efficient and simple manufacturing method for the fabrication of geometric phase elements with femtosecond laser direct writing (FsLDW) and thermal annealing by combining the advantages of high-efficient processing and thermal smoothing effect. In response to the comments, we have emphasized this point in the abstract, introduction and conclusions.

  1. Abbreviations should be defined at first appearance and properly written in same way throughout the manuscript. For eg. FsLDW and FsDLW.

Response:  Thanks. We have revised through the paper.

  1. Fig. 2 caption and the pictures are not clear. Better to show more clear figures along with clear labeling.

Response: Thanks. We have changed the color to white to make it clear. Here is the new figure shown as below.

  1. Page 5, define all the variables used in eqn 3.

Response: Thanks. We have revised.

“Where ρ is the radius of the polar coordinates, λ is the working wavelength of the geometric phase DOE, f is the designed focus length (100 μm).“

Reviewer 3 Report

It is an interesting paper about the fabrication of geometric phase elements by laser writing using an ultra-short pulse source, even if the coupling of laser ablation and post annealing is not so new. The paper is well organized and only minor changes are required prior to publication.

  • The English should be carefully revised.
  • It is not clear why the authors consider their method “high-efficient”. There is no comparison with the performances of other methods.
  • No information about the laser energy was reported in the experimental. In addition, why did the authors choose the laser third harmonic to perform their experiments?
  • About the experimental parameters of the annealing process, it is not clear why the authors changed at the same time both the furnace temperature and the annealing time. The change of only one parameter at a time should be more logical.  

Author Response

  1. It is not clear why the authors consider their method “high-efficient”. There is no comparison with the performances of other methods.

Response: Thanks for the comments. This method lies on fast processing speed, two steps, and low requirement of process conditions (non-vacuum) which are much efficient than other technologies. As we know, it takes at least half an hour to perform vacuum condition for EBL, about 20 mins to spin coating, another 20 mins to remove coating after fabrication and totally 3 hours to obtain the patterns. It is energy-consuming. While for our method, it takes 50 seconds for the fabrication of any pattern in the scale of 200 µm × 200 µm and 1 hour for annealing. Most importantly, all the work are done in the air condition. In response to the comments, the red words have been added.

Paragraph 2, Page 2

“Generally, it takes 50 seconds for the fabrication of any pattern in the scale of 200 µm × 200 µm by FsLDW and 1 hour for the subsequent annealing. Most importantly, the maskless and vacuum-free capabilities make the approach high efficient for the production of geometric phase elements with superior quality”

  1. No information about the laser energy was reported in the experimental. In addition, why did the authors choose the laser third harmonic to perform their experiments?

Response: Thanks. We showed the pulse energy of 4 nJ in paragraph 6 of the text. To make it more clear, we add an illustration in the Materials and Methods.

“All the pulse energy used in the experiments are 4 nJ.”

According to the focus radius defined by r = 0.61λ/NA, a shorter wavelength results in a higher resolution. Herein we used a wavelength of 343 nm achieve higher machining resolution for the nanogrtings with period of 300 nm and nanogroove width of 170 nm.

  1. About the experimental parameters of the annealing process, it is not clear why the authors changed at the same time both the furnace temperature and the annealing time. The change of only one parameter at a time should be more logical.

Response: We mainly want to find and display the best annealing experiment parameters. In the process of searching for the parameters, for example, we found that there is basically no difference for the SEM image for the temperature of 600 oC regardless of the time. Meanwhile, the chosen of temperature and time is much more clearly show the change in the thermal annealing process, indicating that the thermal annealing process has a certain positive effect on the shaping of the gold film.
